# Robotic Biofeedback for Post-Stroke Gait Rehabilitation: A Scoping Review

**DOI:** 10.3390/s22197197

**Published:** 2022-09-22

**Authors:** Cristiana Pinheiro, Joana Figueiredo, João Cerqueira, Cristina P. Santos

**Affiliations:** 1Center for MicroElectroMechanical Systems (CMEMS), University of Minho, 4800-058 Guimarães, Portugal; 2LABBELS-Associate Laboratory, University of Minho, 4800-058 Guimarães, Portugal; 3Life and Health Sciences Research Institute (ICVS), University of Minho, 4710-057 Braga, Portugal; 4Clinical Academic Center (2CA-Braga), Hospital of Braga, 4710-243 Braga, Portugal

**Keywords:** biofeedback mode, biofeedback parameter, human sensing, motor recovery, robotics rehabilitation, sensorimotor augmentation, stroke

## Abstract

This review aims to recommend directions for future research on robotic biofeedback towards prompt post-stroke gait rehabilitation by investigating the technical and clinical specifications of biofeedback systems (BSs), including the complementary use with assistive devices and/or physiotherapist-oriented cues. A literature search was conducted from January 2019 to September 2022 on Cochrane, Embase, PubMed, PEDro, Scopus, and Web of Science databases. Data regarding technical (sensors, biofeedback parameters, actuators, control strategies, assistive devices, physiotherapist-oriented cues) and clinical (participants’ characteristics, protocols, outcome measures, BSs’ effects) specifications of BSs were extracted from the relevant studies. A total of 31 studies were reviewed, which included 660 stroke survivors. Most studies reported visual biofeedback driven according to the comparison between real-time kinetic or spatiotemporal data from wearable sensors and a threshold. Most studies achieved statistically significant improvements on sensor-based and clinical outcomes between at least two evaluation time points. Future research should study the effectiveness of using multiple wearable sensors and actuators to provide personalized biofeedback to users with multiple sensorimotor deficits. There is space to explore BSs complementing different assistive devices and physiotherapist-oriented cues according to their needs. There is a lack of randomized-controlled studies to explore post-stroke stage, mental and sensory effects of BSs.

## 1. Introduction

Gait disabilities, mainly caused by strokes, compromise daily independence, quality of life, professional and social inclusion, and increase the risk of falling in adults [1,2]. Stroke survivors may regain their quality of life through neuroplasticity phenom, elicited by biofeedback systems (BSs) [3,4,5].

In the context of this review, a BS is a robotic device that measures gait-related unconscious parameters through sensors and feedback in real-time this information to users through visual, auditory, or haptic cues, using appropriate actuators [6]. Therefore, patients are aware of their abnormal behaviour, and they are intensively and repetitively encouraged to self-control it (fostering neuroplasticity) towards recovery.

A review on BSs designed for post-stroke gait rehabilitation is essential to find evidence on the BSs’ rehabilitation effects and to identify the challenges and the limitations to be tackled in future research. The most recent reviews on this topic [7,8,9,10] revealed the promising impact of biofeedback on motor recovery. Linda van Gelder et al. [7] suggested future directions on biofeedback research to improve gait function, although they had considered a large variety of participant groups (healthy, runners, stroke/hemiplegia, Parkinson’s, incomplete spinal cord injuries, cerebral palsy, multiple sclerosis, amputees, diabetics, and knee injuries). Similarly, Thomas Bowman et al. [9] analyzed the BSs’ effects in heterogeneous participants, including patients with Parkinson’s disease, stroke, and mild cognitive impairment. It is important to note that the unique post-stroke sensorimotor deficits suggest that a review on the clinical effects of BSs’ merits should be pathology-specific rather than crossing different pathologies. However, the reviews [7,9] do not focus on post-stroke participants, who are the target users for several BSs developed for gait training [7,8,10,11]. On the other hand, Rosalyn Stanton et al. [8] investigated the efficacy of biofeedback to improve performance in lower limb activities (namely, sitting, sit to stand, standing, and walking) after stroke compared to conventional therapy. This review was not specific for gait rehabilitation. Since gait recovery, due to its complexity, requires balance and coordinated activation of muscles, a review focusing on gait rehabilitation is valuable [12]. Jacob Spencer et al. [10] reviewed current evidence and future research directions related to post-stroke gait biofeedback, but they excluded studies involving biofeedback in adjunction with robotic assistive gait training (also not found in [8,9]). However, biofeedback in adjunction with assistive gait training can encourage patients’ active participation, preventing motor dependence on assistive devices [13].

Therefore, there is a need to conduct a scoping review and appoint future research directions on the design of BSs exclusively related to post-stroke gait rehabilitation, including the adjunctive use with assistive devices such as exoskeletons. Moreover, physiotherapist-oriented sensory cues provided according to biofeedback parameters should be innovatively studied once the physiotherapist’s involvement can assure effective use of the robotic devices and foster the patient’s motivation [14].

This study reviewed the technical and clinical specifications of BSs developed for post-stroke gait rehabilitation, including the complementary use with assistive devices and/or physiotherapist-oriented cues, advancing the current literature [7,8,9,10]. This work aims to review the sensors, actuators, and control strategies used to drive the sensory cues according to the sensor’s output (technical specifications); and clinical protocols and evidence concerning the system’s efficacy on post-stroke recovery (clinical specifications). Further, it recommends directions for future research regarding the design of both BSs and clinical studies towards effective and prompt post-stroke gait rehabilitation. The following question was investigated: Which are the technical and clinical specifications of BSs that should be followed in future research to achieve efficacy in post-stroke rehabilitation?

## 2. Materials and Methods

### 2.1. Data Sources and Search Strategy

The studies presented in this review were searched in the following six databases: Cochrane (source: “all except PubMed and Embase”), Embase, PubMed, PEDro, Scopus, and Web of Science, using search field “Title Abstract Keyword” from January 2019 to September 2022. For the search strategy, we used the following keywords: “(biofeedback OR neurofeedback OR (EMG AND feedback)) AND (stroke OR post-stroke) AND (gait OR walk*) AND (rehabilitation OR recovery)”. Studies published in the English language were filtered to those between 2015 and 2021. The reference lists of the relevant studies and reviews were checked.

### 2.2. Study Selection

The studies were checked and selected by the primary researcher (C.P.) according to the following inclusion criteria: original studies; identification of the used technology in the BS; clinical intervention with post-stroke patients; BS designed for gait rehabilitation; clinical assessment of motor effects. The following exclusion criteria were applied: not biofeedback; no study of motor effects in post-stroke; not gait rehabilitation; feedback was not in real-time; duplicate commercial BS; not post-stroke participants.

### 2.3. Data Extraction and Synthesis

The primary researcher (C.P) extracted the relevant specifications of BSs from all eligible studies. A specification was considered relevant when mentioned in more than one study. A second researcher (J.F.) checked the reliability of data extraction. A third researcher (C.S.) allowed a consensus between researchers. Data regarding technical and clinical specifications were extracted from the relevant studies. Technical specifications are the following: sensors used to provide biofeedback and biofeedback parameters; actuators to provide sensory cues; control strategies to manage sensory cues according to sensor’s measures; assistive devices used in adjunction with BSs; physiotherapist-oriented sensory cues controlled according to biofeedback parameters. Clinical specifications address the following: post-stroke participants’ characteristics; clinical protocols employed for BSs’ validation and clinical intervention; sensor-based and clinical outcome measures used to assess BSs’ effects on post-stroke recovery; BSs’ effects on post-stroke recovery. Data extracted were summarized by counting the number of studies in each category for each specification. For example, regarding the specification of sensors used in the BSs, the number of studies using each sensor was counted, grouping the studies using the same sensor.

### 2.4. Methodological Quality Assessment

The quality assessment of the reviewed studies was performed through the Effective Public Health Practice Project (EPHPP) tool for quantitative studies [15]. This tool allows the quality assessment of mixed-methods studies, evaluating the following components as “strong”, “moderate” or “weak”: “selection bias”, “study design”, “cofounders”, “blinding”, “data collection methods”, and “withdrawals and drop-outs”. The global rating is considered “strong” if no component is “weak”, “moderate” if only one component is “weak”, or “weak” if two or more components are “weak”.

## 3. Results

The search strategy (Figure 1) yielded 305 studies, resulting in 206 studies after duplicates’ removal. Then, 185 studies’ titles and abstracts were screened for relevance, and 47 studies were full-text assessed for eligibility, excluding 138 and 16 studies, respectively. A total of 31 studies were included in this review.

### 3.1. Technical Specifications

Table 1 details the technical specifications (sensors and related biofeedback parameters, actuators and related sensory cues, adjunctive assistive devices, physiotherapist-oriented sensory cues) of the reviewed studies.

#### 3.1.1. Sensors and Biofeedback Parameters

Sensors were used in BSs to measure in real-time biofeedback parameters. Biofeedback parameters addressed neuro-biomechanical data (measured by the sensors) related to motor function in the field of gait rehabilitation. The sensors’ devices, related biofeedback parameters, and the locations where the sensors were placed on the human body (in wearable devices) were reviewed.

Four studies used an electromyography (EMG) system [21,22,25,39] or inertial measurement units (IMUs) [31,40,43,46]. EMG-based systems measured electromyographic signals on tibialis anterior (four studies), quadriceps femoris (two studies), gastrocnemius lateralis, soleus, vastus lateralis, rectus femoris, and biceps femoris muscles. IMUs were usually placed on feet [31,40,46]. Three studies referred force sensors [16,18,19], load sensors [22,26,41], pressure sensors [31,32,35], a camera [27,29,34], an optical motion capture system [23,30,45], or an electroencephalography (EEG) system [20,37,42]. Force sensors were allocated on first (two studies) and fifth metatarsal heads (two studies), toe and heel of feet. Besides the feet (two studies), load and pressure sensors were alternatively placed on an exoskeleton’s knee and hip joints [22] or on a cane [32], respectively. Regarding the EEG, two studies discriminated the following target locations: Oz, Cz, ipsilesional hand motor cortex area, and Broadmann area 10. Two studies mentioned force platforms integrated into treadmills [17,38].

The sensors can be conjugated in the same BS. Only two studies included more than one sensor in the same BS, conjugating pressure sensors with IMUs [31] and magnetic resonance imaging (MRI) with EEG system [37]. Non-wearable sensors were employed in 13 studies, namely force platforms, optical motion capture systems, cameras, strain gauges, balance board, infrared sensors, plantar pressure measurement mat, and MRI system (Table 1).

Different sensors were used to measure the same biofeedback parameter. Kinetic parameters were evaluated in 12 studies by pressure, force, or load sensors, force platforms, pneumatic insoles, plantar pressure measurement mats, or balance boards [16,17,22,26,28,29,31,35,36,38,41] Spatiotemporal parameters were monitored in 12 studies by force sensor, load sensors, strain gauges, IMUs, cameras, optical motion capture systems, or infrared sensors [18,19,24,26,27,29,30,31,33,34,40,46]. Physiological parameters were measured in eight studies by EMG, EEG, fNIRS, or MRI systems [20,21,22,25,37,39,42,44]. Kinematic parameters were assessed in five studies by IMUs or optical motion capture systems [23,31,40,43,45].

#### 3.1.2. Actuators and Biofeedback Mode

According to the sensor’s data, actuators were used in BSs to provide sensory cues (i.e., biofeedback mode) to the users. The actuators’ devices, biofeedback modes, and location where the actuators were placed on the human body (in wearable devices) were reviewed.

The following three biofeedback modes were reported: visual (25 studies), auditory (13 studies), and haptic (4 studies). A total of 11 studies combined different biofeedback modes, conjugating visual with auditory cues (9 studies), auditory with haptic, and visual with haptic.

Visual cues were provided by non-wearable devices as screens in 24 studies, using computers, monitors, televisions, tablets, or projectors; or as lights. Auditory cues were supplied by speakers in 11 studies, headphones [24], earphones [40], or an indicator placed on the waist [32]. Speakers were usually integrated into non-wearable devices such as computers, televisions, mobile phones, or tablets, but they were also placed on the human body, namely the waist and foot. Haptic cues were delivered by vibrators (two studies) placed on the paretic wrist or leg, programmable plates integrated into a robotic assistive device, or an electrotactile system using electrodes placed on the user’s unaffected thigh. The electrotactile system provided haptic cues without stimulating any unintended muscular activity that can alter the patient’s gait [19].

#### 3.1.3. Biofeedback Control Strategies

Biofeedback control strategies are described below, explaining how and when the sensory cues (visual, auditory, and/or haptic cues) were provided to the users according to the real-time data from the sensors.

Visual cues were provided as follows. J Jung et al. [36] exhibited a color map of the feet, setting the color from light blue to red and yellow as the pressure load increased. Byl et al. [31] flashed foot pressing indicators and the values of step length, stride width, and toe-off angle, adding statements on the screen according to them (e.g., “you are doing perfectly”, “twist your feet inwards”, “step further on the left”). Bae et al. [27] showed a motion observation screen of simultaneously a reference model and the real-time video of the participant’s ankle from the side.

Tamburella et al. [22] displayed blue, red, and white colored stripes portioned into stages within the gait cycle representing over-activation, under-activation, and optimal muscle contraction, respectively, according to a targeted reference muscle activation profile [22]. Moreover, weighted averages of the joint human–robot interaction (HRI) torques were exhibited in an array of line graphs (1 line/gait phase) [22]. Day et al. [23], Nagano et al. [30], and Park et al. [45] overlaid a trace/round points representing a one-dimensional summary of performance, MFC, and position of ankle markers during the swing phase, respectively, on a goal zone (horizontal lines in [23,30] and vertical bars in [45]). Ochi et al. [26] and Kim et al. [35] activated lights and modulated a graphical depiction, respectively, according to the real-time load of the user’s weight on each limb during the stance phase.

Relatively to auditory cues, K-S Jung et al. [32] produced a sound whenever the peak vertical force on a cane was higher than a threshold. Choi et al. [41] generated a beep sound when a pre-defined load was detected in the stance phase. Hankinson et al. [43] produced a target beat per minute that was silenced (and then a metronome plays instead) when the participant did not perform a target angle range of movement.

Concerning haptic cues, Ma et al. [16] activated a vibrator at 220 Hz and 1 G when the medial plantar force was less than a threshold (50% lateral plantar force). Afzal et al. [18] enabled a vibrotactile cue at 100% intensity (1.5 G) when the paretic leg’s swing phase started and ended after 200–500 ms (time increased or decreased with the error between desired and actual time symmetry ratio) or the end of the swing phase, whichever came first [18]. On the other hand, the vibration was enabled at 25–100% intensity (intensity increased or decreased with the error) during the paretic leg’s swing phase [18].

Visual and auditory cues were conjugated as follows. Genthe et al. [17] displayed horizontal a green and a black line with a cursor (X) indicating the target and actual AGRF, respectively. Guzik et al. [24] presented feet characters according to the measured step length, a rectangle character indicating the area where the feet should be placed, and the values of recent and target step lengths. Hsu et al. [28], Mihara et al. [44], and Surucu et al. [25] presented vertical bars that change their height to reflect the magnitude of the participant’s weight-bearing on the paretic leg, SMA activation status, and muscle activity, respectively. Arpa et al. [21] and Tsaih et al. [39] demonstrated muscle contraction amplitude over time through line graphs. These studies enabled auditory cues (audible tone [17] and beep sound [28,44]) when the target was achieved.

Givon et al. [29] presented the virtual-reality games from Microsoft Xbox Kinect (as in Song et al. [34]), Nintendo Wii Fit, and SeeMe VR system. Skvortsov et al. [46] slowed down the movement around the virtual environment when an out-of-range stance time is detected. Moreover, a vertical bar (with marked target range) that changes its height to reflect the stance time was presented.

Shin et al. [33] displayed green, blue, or red colors when the deviation of the actual and step length from the target is within 10 cm, ranged from 10 to 30 cm, is more than 30 cm, respectively. Additionally, auditory cues were provided corresponding to the target step cycle initiation [33]. Schlieβman et al. [40] referred to a colored graph comprising consecutively green, yellow, and red ranges selected to match the comparison between calculated parameters and reference values. Furthermore, three auditory instructions were provided using the Android-OS default Google text-to-speech engine: “stance duration correct”, “stance duration shorter”, and “stance duration much shorter” for green, yellow, and red ranges, respectively [40].

Relatively to auditory and haptic cues conjugation, Khoo et al. [19] gave an audio tone at 200 Hz and electrotactile stimulation at 80–250 µs pulse width, 250 Hz, and up to 115 mA (using 500 Ω test load) when the swing time of the affected limb exceeded a threshold [19]. Moreover, the audio tone was activated when the affected leg established a heel-strike and stopped at a pre-defined time [19].

Concerning visual and haptic cues conjugation, Boehm et al. [38] displayed the vertical force distribution on each foot and induced a vibratory cue from the force plates when the users supported most of their weight with the stance foot. From the reviewed studies, we verified that the sensory cues were usually provided to the users following principles of a Finite State Machine (FSM) approach, which compared the user’s current state (based on real-time data from sensors) with a threshold or reference. Details about the controllers (linear, nonlinear, intelligent, adaptive) were not explicitly mentioned. Study [22] did not use and seven studies did not mention a threshold or reference in their biofeedback control strategies. The remaining studies defined a threshold or reference according to the non-paretic limb (six studies), a baseline trial performed prior training without sensory cues (nine studies), data from healthy subjects (six studies), maximum voluntary contraction (two studies) in the case of EMG-based biofeedback, body weight (two studies), and the paretic limb. Regarding the baseline trial approach, the user tried some pre-defined thresholds consecutively, and the difficulty to achieve them was evaluated [17], or the threshold was directly calculated from baseline data. Six studies have updated the threshold during training agreeing to the user’s performance [20,32,37,40,43,46]. User’s performance was usually defined according to the frequency of achieving the threshold or reference in all studies. The time when the sensory cues are enabled depends on the control’s periodicity and the type of reinforcement intended (positive and negative reinforcement to enable the sensory cue at a condition that must be learned or avoided, respectively). The periodicity of the biofeedback control strategies was not mentioned in 14 studies. The remaining 16 studies controlled sensory cues at each gait cycle (14 studies), a fixed time in the case of EEG-based biofeedback (2 studies), or at each third stride. Three studies reported the update of the periodicity during training, usually according to the user’s performance [17,39,40]. Positive and negative reinforcement was provided in 24 and 13 studies, respectively.

#### 3.1.4. Assistive Devices Used in Adjunction with BSs

Assistive devices were used in adjunction with BSs to accelerate gait rehabilitation once they provided user-oriented intensive and repetitive training. A total of 13 studies reported the complementary use of assistive devices with BSs, namely, treadmills (11 studies), a cane, hip and knee exoskeleton, ankle-foot orthosis, gait-assistance robot with four robotic arms to control both thighs and legs, EMG-Functional Electrical Stimulation (FES) system targeting tibialis anterior muscle, kinetic immersive interface for neuromuscular coordination enhancement (KIINCE), and a cable-driven robotic system to correct force on the pelvis in the lateral direction towards the paretic side. Four studies assisted with the treadmill and another assistive device in adjunction with BSs, that was, the hip and knee exoskeleton [22], the ankle-foot orthosis [45], the gait-assistance robot [26], and the cable-driven robotic system [28]. In total, 14 studies included non-wearable assistive devices, and the remaining study did not mention wearability.

#### 3.1.5. Physiotherapist-Oriented Sensory Cues

Only four studies stated physiotherapist-oriented sensory cues, being all visual ones. Physiotherapist-oriented sensory cues were provided along with patient-oriented cues, allowing physiotherapists to follow the biofeedback training. Consequently, they supplied adequate instructions to the patients combining their expert visual inspection with objective data from sensors, aiming to accelerate the recovery. A total of 26 studies did not refer to the assessment of the sensory cues by physiotherapists (NM in Table 1) but implemented patient-oriented cues that physiotherapists can simultaneously assess due to the nature of the actuators’ device, namely, screen, light, speaker, indicator, or programmable plate. Five studies did not allow physiotherapists to assess patient-oriented cues (NA in Table 1) because of the nature of the actuators’ device, specifically, headphone, earphone, vibrator, and/or electrotactile system.

### 3.2. Clinical Specifications

Table 2 summarizes the clinical specifications (sample size, study design, training dosing, and evaluation time points) found in the reviewed studies.

#### 3.2.1. Post-Stroke Participants

The review englobes a sample size of 660 post-stroke participants, with 17 post-stroke participants being the median sample size of the reviewed studies. The maximum and minimum sample sizes were found in Guzik et al. and Nan et al. [20,24] with 50 and 2 post-stroke subjects, respectively.

The post-stroke participants are mostly characterized by age (30 studies), gender (30 studies) [36,37,38,39,40,41,42,44,45,46], time post stroke (27 studies) [16,17,18,21,22,23,24,25,26,27,28,29,30,31,32,33,34,36,37,38,39,40,41,42,44,45,46], hemiplegic side (26 studies) [16,17,18,21,22,23,24,25,26,27,30,31,32,33,34,35,36,37,38,39,40,41,42,44,45,46], stroke aetiology (18 studies) [16,18,20,22,25,26,27,29,30,32,33,35,36,37,38,40,41,42], and body mass (15 studies) [16,18,20,22,27,28,30,32,33,34,36,38,39,41,46].

#### 3.2.2. Study Design

The review includes 15 uncontrolled, 11 randomized controlled, 4 non-randomized controlled, and 1 randomized cross-over study (Table 2). A total of 10 controlled studies were balanced on the number of participants between experimental and control groups (Table 2).

The clinical protocols of uncontrolled studies were based on training and control or baseline procedures. The training procedure allowed the users to follow sensory cues provided by the BS towards recovery. The control procedure was similar to the training one, except for the use of biofeedback. Therefore, the control procedure is used to control the biofeedback effects within the participant. Additionally, the control procedure was used to specify training thresholds.

Five uncontrolled studies only performed training and control procedures, randomly assigning the order between participants to limit carry-over, practice, and order effect [16,18,33,37,38]. Moreover, two studies implemented a wash-out period (i.e., a period without biofeedback; 1 h [33] and 1 month [37] considering a training dosing of 5 min and twice a week over 2 months for 50 min, respectively) between control and training procedures to exclude the carry-over, practice, and order effect.

Three uncontrolled studies complemented biofeedback training with conventional physiotherapy exercises [24,37] or robotic assistive devices [28,38]. Hsu et al. performed training with biofeedback only and training by combining pelvic corrective force and biofeedback, randomly assigning the order between participants [28]. The control procedure did not include any type of biofeedback and robotic assistive device [28]. In opposition to Hsu et al., Boehm et al. implemented a control procedure without the biofeedback but with the robotic assistive device [38].

The cross-over study from Tamburella et al. randomly created two groups of participants to perform cross-over training with two BSs complemented with robotic assistive devices and conventional rehabilitation [22]. The control procedure was executed with the robotic assistive device [22].

Controlled studies included two groups of participants who executed specific interventions: experimental and control groups. Experimental groups trained with the biofeedback system. Additionally, experimental groups accompanied biofeedback training with conventional rehabilitation exercises (12 studies) [19,21,25,26,27,31,32,35,39,41,43,44] and robotic assistive devices (2 studies) [26,27]. Control groups usually did not perform biofeedback training, only addressing conventional rehabilitation. However, the control groups from Song et al. followed distinct biofeedback from the experimental group [34], and Lee et al. and Mihara et al. implemented sham neurofeedback [42,44]. Bae et al. and Ochi et al. included and did not include, respectively, the robotic assistive devices during control intervention [26,27].

#### 3.2.3. Clinical Protocols

This section reviews the existence and duration of the familiarization prior to the training, the motor task executed during familiarization and biofeedback training, the assistance provided by the assistive devices (in case of their use), and the training dosing.

Three studies mentioned familiarization of the BS prior to training (10 min in [16], 2 min in [23], the value was not mentioned in [18]) while study [28] performed familiarization during training. Regarding the motor task, 20 studies addressed walking (10 studies walking on treadmill and the remaining walking on level-ground) while the remaining tackled sitting (8 studies) [20,21,25,27,37,39,42,44] or standing (2 studies) [29,34]. Training during sitting was performed using EMG-feedback, neurofeedback (motor imagery), and biofeedback complemented with EMG-FES. Virtual-reality games included standing training. For gait training, only two studies did not implement a self-selected walking speed [22,26].

Some studies evaluated the effects of biofeedback training during motor assistance driven by assistive devices. The exoskeleton from Tamburella et al. applied body-weight support equal to 50% of the participant’s body weight, 100% guidance assistance, and 1.3 km/h speed [22]. The gait-assistance robot from Ochi et al. was predominantly used in active-assistive mode reaching a 0.75 km/h speed, 10–30° hip ROM, and 5–10° ankle ROM [26]. The EMG-FES system from Bae et al. stimulated 1–50 mA by 0.1 s rise, 5 s on, and 2 s decay according to the subject’s ankle joint [27]. The cable-driven robotic system from Hsu et al. employed a pelvic corrective force equal to 9% of participants’ body weight at the timing of heel strike of the paretic leg and lasted for 400 ms [28]. The KIINCE from Boehm et al. moved its plates forward and afterward out of phase in a simplified walking motion [38]. The treadmill speed from Guzik et al. increased by 5 to 10% during consecutive training sessions [24].

The training dosing usually implied multiple sessions with a fixed frequency per week. The maximum, minimum, and median training dosing were twice a week for 12 weeks at 60 min/session in Givon et al., two 15 min sessions in Day et al., and three times a week for 4 weeks at 30 min/session in Kim et al., respectively. Only five studies, all uncontrolled ones, completed the training procedure on a single day.

#### 3.2.4. Sensor-Based and Clinical Outcomes

The evaluation time points were mostly pre- and post-training (31 studies) to assess sensor-based (29 studies) and clinical (22 studies) outcomes (Table 3). Retention evaluation time points were executed immediately after training (immediate post-training) and follow-up in 28 and 9 studies, respectively. The later, earlier, and usual follow-up evaluations were performed 3 months [21,29], 2 min [17], and 1 month [21,30,37,40] post-training (Table 2), respectively.

The evaluated sensor-based outcomes were categorized on spatiotemporal (18 studies), kinetic (13 studies), kinematic (10 studies), and physiological (8 studies) outcomes.S patiotemporal outcomes mostly included step length (8 studies) and walking speed (9 studies). Mostly addressed kinetic outcomes were plantar pressure (2 studies), peak vertical force (2 studies), peak joint torque (2 studies), and grip strength (2 studies). Joint ROM was the most evaluated kinematic outcome (5 studies) [21,23,25,31]. Physiological ones usually included EMG- (5 studies) and EEG-based (3 studies) outcomes. EMG-based outcome was muscle activation. EEG-based outcomes comprised relative individual alpha-band (IAB) amplitude, sensorimotor rhythm (SMR) waves amplitude, and alpha-band weighted node degree of functional connectivity.

Clinical outcomes addressed post-stroke stage (7 studies), motor (19 studies), mental (8 studies), and sensory (2 studies) effects on post-stroke participants. Post-stroke stage was mostly evaluated by Fugl–Meyer (FM) (6 studies). Motor effects were mostly assessed by 10 m walk test (12 studies) and Timed Up and Go (TUG) (13 studies). Mental effects were determined by Quebec User Evaluation of Satisfaction with Assistive Technology 2.0 (QUEST 2.0) (2 studies), Visual Analogue Scale (VAS) (2 studies) of motivation, mood, satisfaction, and pain, and Likert-type satisfaction scale (2 studies). Sensory effects were revealed by proprioception test and Pin Prick (PP) test for haptic effects and by Start Cancellation Test for visual effects.

#### 3.2.5. BS Effects on Post-Stroke Recovery

Table 3 shows the BS effects on post-stroke recovery based on the parametric and non-parametric statistical tests performed by the reviewed studies to conclude the existence of statistically significant differences (alpha set to 0.05) on the measured outcomes between the evaluation time points on both experimental and control groups.

Regarding sensor-based outcomes, 15, 12, 8, and 5 studies reached statistically significant improvements on spatiotemporal, kinetic, kinematic, and physiological outcomes, respectively, between at least two evaluation time points. Studies from Givon et al., Nagano et al., Hsu et al., Skvortzov et al., Park et al., and Mottaz et al. [28,29,30,37,45,46] did not find statistically significant improvements between all evaluation time points. Nan et al. [20] did not evaluate physiological outcomes using statistical tests. Moreover, 8, 6, 6, and 2 controlled studies exhibited higher statistically significant improvements on the experimental group than the control group concerning spatiotemporal, kinetic, kinematic, and physiological outcomes, respectively.

For the clinical outcomes, four, five, and four studies achieved statistically significant improvements on post-stroke stage, motor, and mental function, respectively, between at least two evaluation time points. Two, two, four, and two studies did not evaluate post-stroke stage, motor, mental, and sensory effects, respectively, using statistical tests. Givon et al., Hankinson et al., and Schlieβmann et al. [29,40,43] did not find statistically significant improvements on clinical outcomes between all evaluation time points. Additionally, three, five, and two controlled studies demonstrated higher significant improvements on the experimental group than the control group concerning post-stroke stage, motor, and mental effects, respectively.

### 3.3. Methodological Quality Assessment

Table 4 shows the results of the EPHPP tool. The “selection bias” is “weak” (15 studies) or “moderate” (13 studies) in most studies. The “study design” is “strong” or “moderate” in 16 and 15 studies, respectively. The “confounders”, “blinding”, and “withdrawals and drop-outs” components are “strong” in most studies (29, 24, and 21 studies, respectively), being “weak” in 2, 7, and 10 studies, respectively. The “data collection methods” are “strong” in all studies. The global rating is “strong”, “moderate”, or “weak” in 5, 12, and 14 studies, respectively.

## 4. Discussion

This review aims to investigate the technical and clinical specifications of BSs for gait rehabilitation that should be followed in future research to achieve efficacy for post-stroke recovery. Despite the current promising research, more scientific evidence is needed to rate the efficacy of biofeedback accurately [8]. The research question is answered in the following sections.

### 4.1. Technical Specifications

There was no sensor technology found that was common to most studies, as appointed in [9]. In of spite that, sensors that measure kinetic and spatiotemporal parameters were usually employed in the reviewed studies. Pressure sensors and force platforms evaluated kinetic biofeedback parameters, and force and load sensors additionally measured spatiotemporal biofeedback parameters. IMUs and motion capture systems determined kinematic and spatiotemporal biofeedback parameters. Cameras assessed the body’s movement. EMG and EEG systems measured muscular activation from the tibialis anterior and quadriceps femoris muscles and alpha-band EEG signals, respectively.

Even though most of the studies implied only one sensor on the BS, Byl et al. and Mottaz et al. [31,37] combined pressure sensors and MRI with IMUs and EEG, respectively. As reviewed in Byl et al. and Mottaz et al. [31,37], future research should study the effectiveness of combining multiple biomechanical and physiological sensors to personalize biofeedback for post-stroke users given their variable motor deficits. Further, the use of multimodal sensors may enable a more holistic BS-based gait training and assessment, attending to intra- and inter-subject motor variability [7,10].

Regarding sensors’ wearability, wearable sensors were used in most studies. Wearability allows unique assessments of body motion during ambulatory training in a non-fixed facility [10,47]. Thus, allowing the users to practice in multiple spaces, encouraging training dosing to increase in everyday scenarios and, consequently, accelerating recovery [7,48]. However, EMG and EEG sensors required a time-consuming preparation in opposition to IMUs, pressure, force, and load sensors that were fast positioned on the feet. This finding guides future research to select wearable sensors with fast positioning.

Concerning biofeedback mode, most studies reported visual biofeedback mode, as concluded in [9], using screens from monitors, televisions, tablets, or projectors. Auditory and haptic biofeedback modes were usually applied using speakers integrated into computers, televisions, or tablets, and vibrators, respectively. Even most studies have implied one mode, visual and auditory cues were combined once multimodal biofeedback can reduce the user’s cognitive load compared to a single-mode [7]. Therefore, there is space to explore auditory and haptic modes, as retained in [7,10], so that post-stroke users with multiple sensorial deficits can take advantage of multimodal biofeedback rehabilitation. In this manner, the physiotherapist has the necessary resources to personalize the training according to the patient’s imminent sensorial deficits.

Regarding actuators wearability, only haptic biofeedback was provided using wearable actuators in most studies, allowing ambulatory practice on daily-like scenarios as overground walking. Thus, future research should study the impact of using wearable actuators as earphones and augmented reality glasses for auditory and visual modes, respectively, benchmarking the results with non-wearable solutions. Haptic actuators were placed on both upper or lower limbs, fostering the conclusion that haptic feedback on the body, either at or away from the desired body segment to be changed, can improve motor performance [48].

Most biofeedback control strategies compared sensor data with a threshold or reference obtained from the user’s limbs, a baseline trial, maximum voluntary contraction, body weight, or data from healthy subjects, having a need to benchmark these methods. Regarding periodicity, most BSs for gait training have controlled the sensory cues at each gait cycle. Studies exploring EMG- and EEG-based biofeedback, performed during sitting, did not mention the control’s periodicity or attended a fixed-time control, respectively. Future studies should clearly state the control’s details (type, quality, and periodicity), as concluded in [7], fostering the research’s reproducibility. Less than half of the studies updated the threshold or reference and the control’s periodicity during training agreeing with the time for achieving the threshold or reference, intended as the user’s imminent disability level. In this sense, the rehabilitation was personalized according to the user’s imminent disability level, avoiding frustration and dependence on the sensory cues, respectively [37,49]. However, there is space to continue the investigation of biofeedback control strategies’ personalization according to the user’s imminent disability level, reporting significant evidence of this adaptation.

Positive reinforcement was employed in most biofeedback control strategies, as referred in the systematic review of Bowman et al. [9], drawing the user’s attention to a condition that should be learned and repeated. Even most studies with multiple biofeedback modes applied positive reinforcement on both. In this manner, there is space to explore if combining both positive and negaztive reinforcements, taking into account the cognitive effort, accelerates the relearning process by aware the patients of motor conditions that must be repeated and avoided, respectively.

Visual cues were usually active during training, which can lead to visual reliance and high cognitive effort [50,51]. They provided detailed biofeedback through graphs and scenarios modulated on shape, color, and size according to the control strategy. Auditory and haptic cues were usually inactive, being enabled according to the control strategy at a fixed intensity once intensity adaptation may not be perceived by post-stroke users due to their sensory deficits [52].

Non-wearable assistive devices, mostly treadmills, were used in adjunction with BSs. Treadmills encourage an intensive practice of walking at a controlled and stable gait speed [11]. Robotic assistive devices such as exoskeletons, robotic arms, FES systems, robotic platforms, or cable-driven robotic systems were also applied. They were usually controlled to provide 100% guidance assistance. These robotic systems intensively and repetitively assisted patients on motor tasks while BSs fostered active participation during training [22]. The future directions should continue exploring the contribution of BSs, as an alternative and complementary medicine approach, behind different closed-loop controlled wearable assistive devices.

Physiotherapists guide and instruct the patients according to their specialized knowledge towards recovery [53]. Additionally, physiotherapists’ intervention during training allows the safe and effective use of BSs and robotic assistive devices [6]. Although it was not stated in most studies, BSs could provide objective real-time information about the patients’ motor behaviour to the physiotherapists during the training, complementing visual inspection of the patients, as in [22,25,31,42]. Moreover, BSs could fulfil the lack of physical contact between the physiotherapists and patients during training with robotic assistive devices [54]. Future research should explore the design of physiotherapist-oriented sensory cues according to the physiotherapist’s needs to enrich their contribution to rehabilitation.

### 4.2. Clinical Specifications

Clinical studies related to the effects of BSs on post-stroke recovery had been carried out with a median sample size of 17 post-stroke participants, serving as a reference for future research. Post-stroke participants were selected and characterized mostly by age, gender, time post-stroke, hemiplegic side, stroke aetiology, and body mass, avoiding influencing the research evidence. Future clinical studies may include quantitative characterization of participants’ disability level before the intervention, increasing the reliability of their conclusions.

Literature includes randomized balanced controlled and uncontrolled studies as high-quality and proof of concept research designs, respectively, addressing a median training dose of three times a week for 4 weeks at 30 min/session. Familiarization was typically not stated. However, future research should appoint the existence and duration of this procedure once it can influence the research reproducibility. Moreover, further randomized controlled studies need to be conducted to find clinical evidence regarding the efficacy of biofeedback on post-stroke motor recovery; and potentiate benchmarking of biofeedback technologies and the standardization of their technical and clinical specifications [7,9,10].

Biofeedback training usually involved walking at a self-selected speed, but neurofeedback and biofeedback complemented with EMG-FES were trained during sitting and virtual reality games during standing. Evaluation time points occurred at pre-training and post-training in most studies, evaluating the effects of BSs on post-stroke recovery considering sensor-based and clinical outcomes. Retention procedures were also performed to evaluate motor learning immediately (in most studies) and follow-up post-training. Follow-up evaluations usually occurred 1 month post-training to assess the long-term benefits of the intervention.

In comparison with kinematic and physiological outcomes, spatiotemporal and kinetic ones were the most evaluated sensor-based outcomes. It was expected that most studies would apply kinetic or spatiotemporal biofeedback parameters. Step length and walking speed were highlighted between spatiotemporal outcomes. Clinical outcomes assessed motor effects in most of the studies using the 10 m walk test and TUG. Less than half of the studies also measured the post-stroke stage, mental and sensory effects through clinical outcomes. BSs exert direct action on cognitive and sensory functions once the patients are encouraged to self-control their motor behavior according to the coding scheme of the sensory inputs [13]. Therefore, future studies should fulfil this gap by conducting a user-specific holistic assessment, also including kinematic and physiological sensor-based outcomes, post-stroke stage, mental and sensory clinical effects.

Most studies achieved statistically significant improvements on at least one spatiotemporal, kinetic, kinematic, and physiological sensor-based outcomes, post-stroke stage, motor and mental clinical effects between at least two evaluation time points. Although most studies implied kinetic or spatiotemporal biofeedback parameters, they obtained promising results concerning kinematic and physiological effects and clinical-based outcomes. Moreover, most controlled studies exhibited higher statistically significant improvements on the experimental group than the control group concerning spatiotemporal, kinetic, and physiological sensor-based outcomes, motor and mental clinical effects. In this manner, these positive trends indicate the promise of the efficacy of biofeedback on rehabilitation, as concluded in [7,8,9,10]. Future research should evaluate clinical sensory effects using statistical tests to power the research conclusions.

### 4.3. Future Directions and Challenges to Overcome

There is space to explore: (i) the effectiveness of combining multimodal wearable sensors to cope with variable sensorimotor deficits of stroke survivors; (ii) auditory and haptic biofeedback modes so that post-stroke patients who exhibit multiple sensorial deficits can take advantage of biofeedback rehabilitation; (iii) the impact of using wearable actuators such as earphones and augmented reality glasses, benchmarking the results with the standard non-wearable solutions; (iv) the contribution of BSs behind different robotic assistive devices; (v) the design of physiotherapist-oriented sensory cues according to the physiotherapist’s needs to enhance the physiotherapist–patient interaction; and, (vi) the effects of biofeedback, using statistical tests to power the research conclusions, on kinematic and physiological sensor-based outcomes, post-stroke stage, mental and sensory clinical outcomes. Moreover, future studies on robotic biofeedback for post-stroke gait rehabilitation should: (i) clearly state the biofeedback control’s details (type, quality, and periodicity) and appoint the existence and duration of the familiarization procedure to foster the research’s reproducibility and benchmarking; (ii) include quantitative characterization of participants’ disability level before the intervention to increase the reliability of the conclusions; (iii) biofeedback control strategies’ personalized according to the user’s imminent disability level and the combination of positive and negative reinforcement to accelerate the relearning process; and, (iv) conduct randomized controlled studies to find high-quality clinical evidence regarding the efficacy of biofeedback on post-stroke motor recovery and potentiate benchmarking of biofeedback technologies.

### 4.4. Study Limitations

The main limitation of this review is the moderate or weak quality in most of the reviewed studies mainly due to the rate of the components “selection bias” and “withdrawals and dropouts” (Table 4). Future research should focus on planning studies with strong quality to power their results and conclusions, as concluded in [7,9].

## 5. Conclusions

Most BSs measured kinetic and spatiotemporal parameters using wearable sensors and compared these real-time data with a reference or threshold to control non-wearable actuators capable of providing visual biofeedback following a positive reinforcement. Clinical protocols were executed with a median of 15 post-stroke participants performing walking training three times a week for 4 weeks at 30 min/session. Evaluation of spatiotemporal and kinetic sensor-based outcomes and clinical motor effects were usually performed at pre-training, post-training, and 1-month follow-up. Most controlled studies exhibited higher statistically significant improvements on the experimental group than the control group.

Future studies should report the control’s periodicity and familiarization procedures and explore kinematic and physiological sensor-based outcomes, post-stroke stage, mental and sensory clinical effects. There is space to explore the effectiveness of using multiple wearable sensors and actuators to provide biofeedback; the effects of combining both positive and negative reinforcement on biofeedback control strategies; the contribution of BSs behind different robotic assistive devices; physiotherapist-oriented sensory cues according to the physiotherapist’s needs; and the personalization of biofeedback training according to the user’s imminent disability level.

## Figures and Tables

**Figure 1 sensors-22-07197-f001:**
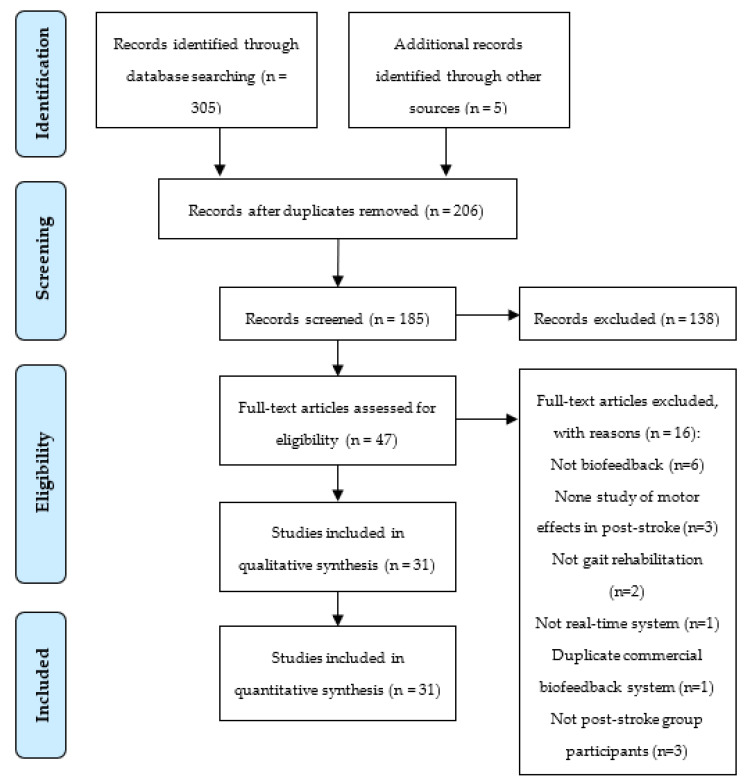
Flow diagram of search strategy based on PRISMA.

**Table 1 sensors-22-07197-t001:** Technical specifications (NA—Not Applied, NM—Not Mentioned, PR—Positive Reinforcement, NR—Negative Reinforcement).

Studies (Year)	Sensor			Actuator			Assistive Device	Therapist-Oriented Cue	Control (Threshold Source, Periodicity, Reinforcement)
	Device	Biofeedback Parameter	Location	Device	Mode	Location			
Ma et al. (2018) [16]	2 thin-film force sensitive resistors (FSRs) (A301, Tekscan Co., Ltd., Wood Dale IL, USA)	medial and lateral plantar forces	Paretic first and fifth metatarsal heads	1 vibrator (XY-B1027-DX, Xiongying electronics Co., Ltd., China)	haptic	paretic wrist	NA	NA	paretic limb, gait cycle, PR
Genthe et al. (2018) [17]	force platforms (Bertec Corporation, Columbus, OH, USA)	anterior–posterior ground reaction force	dual-belt treadmill	screen and speaker (MotionMonitor, Illinois, USA)	visual, auditory	non-wearable	treadmill	NM	baseline, gait cycle, PR
Afzal et al. (2019) [18]	4 FSRs (Tekscan, A401)	time symmetry ratio	toe, metatarsal 1, metatarsal 5, and heel of feet	vibrotactor array (6 units, Precision Microdrives 310-101)	haptic	paretic leg	NA	NA	healthy, gait cycle, PR and NR
Khoo et al. (2017) [19]	6 FSRs (TekScan)	heel-strike and toe-off events, stance and swing times	3 at the front towards the toe and 3 at the back towards the heel of feet	piezo speaker (NM), electrotactile system	auditory, haptic	waist, electrode placed on the user’s thigh on the unaffected side	NA	NM, NA	non-paretic limb, gait cycle, NR
Nan et al. (2019) [20]	EEG system (Compact 823, Meditron, Electromedicina Ltd.a, São Paulo, Brazil)	relative individual alpha band amplitude in the target location (Cz, Oz)	head	computer screen (NM)	visual	non-wearable	NA	NM	baseline, 2 s, PR
Arpa et al. (2019) [21]	EMG system (Neurotrac ETS Simplex 2005)	EMG signal	tibialis anterior and quadriceps femoris muscles	computer monitor, speaker (NM)	visual, auditory	non-wearable	NA	NM	maximum voluntary contraction, NM, PR
Tamburella et al. (2019) [22]	BS1: EMG system (g.tec, Austria); BS2: load cells (Lokomat)	BS1: EMG signal; BS2: weighted averages of the joint human–robot interaction torque	BS1: tibialis anterior, gastrocnemius lateralis, soleus, vastus lateralis, rectus femoris, biceps femoris of the affected leg; BS2: hip and knee joints of exoskeleton	BS1: computer screen (NM); BS2: screen (NM)	visual	non-wearable	treadmill, hip and knee exoskeleton (Lokomat)	BS1: NM; BS2: visual	BS1: healthy, gait cycle, PR and NR; BS2: NA, gait cycle
Day et al. (2019) [23]	Optotrak Certus optical motion capture system (Northern Digital, Waterloo, ON, Canada)	information from sagittal plane hip and knee angles was condensed to a one-dimensional summary of performance	markers attached on bilateral lower limbs	tv screen (NM)	visual	non-wearable	treadmill (Woodway, WI)	NM	healthy, gait cycle, PR and NR
Guzik et al. (2020) [24]	strain gauge (NM)	step length	treadmill	screen, headphones (NM)	visual, auditory	non-wearable	Gait trainer 2 treadmill (Biodex)	NM, NA	NM, gait cycle, PR
Surucu et al. (2021) [25]	EMG system (Electronica Pagani Italy Modular Biofeedback)	EMG signal	tibialis anterior muscle	computer screen, speaker (NM)	visual, auditory	non-wearable	NA	visual	NM, PR
Ochi et al. (2015) [26]	load sensors (NM)	stance phase and load amount	inserted between the sole of the foot and the foot bed of the shoe (feet)	lights (NM)	visual	non-wearable	gait-assistance robot (4 robotic arms control both thighs and legs), treadmill	NM	body weight, gait cycle, NM
Bae et al. (2020) [27]	web cam (NM)	participants’ ankle from the side	non-wearable	tv screen (NM)	visual	non-wearable	EMG-FES (tibialis anterior muscle)	NM	healthy, NM, PR and NR
Hsu et al. (2019) [28]	pneumatic insole (NM)	vertical force	under feet	tv screen (NM)	visual	non-wearable	treadmill (Woodway, WI), cable-driven robotic system (corrective force on pelvis)	NM	body weight, gait cycle, PR
Givon et al. (2016) [29]	BS1 and BS3: camera (Microsoft Xbox Kinect, SeeMe VR system); BS2: balance board (Nintendo Wii Fit)	BS1 and BS3: body movement; BS2: centre of pressure	non-wearable	tv screen, speakers (NM)	visual, auditory	non-wearable	NA	NM	NM
Nagano et al. (2020) [30]	Optotrak Certus optical motion capture system (Northern Digital, ON)	minimum foot clearance	one marker on the toe of the affected limb	monitor (NM)	visual	non-wearable	treadmill	NM	baseline, gait cycle, PR and NR
Byl et al. (2015) [31]	barometric pressure sensors (NM); IMUs (accelerometer, magnetometer, gyroscope; NM)	ground reaction forces (foot pressing indicators); step lengths, stride widths, and toe-out angles	toe, the first and second metatarsophalangeal joint, the fourth and fifth metatarsophalangeal joint, and the heel of feet; feet	iPad screen (Apple, USA)	visual	non-wearable	treadmill	visual	NM
K-S Jung et al. (2020) [32]	pressure sensor (NM)	peak vertical force	cane	indicator (NM)	auditory	waist	cane	NM	baseline, NM, PR
Shin et al. (2017) [33]	infrared sensors (NM)	step length, step cycle initiation	on the rail and both sides of treadmill	tv screen, speakers (NM)	visual, auditory	non-wearable	treadmill (Motorika Reoambulator)	NM	non-paretic limb, gait cycle, PR and NR
Song et al. (2015) [34]	camera (Microsoft Xbox Kinect)	body movement	non-wearable	tv screen, speakers (NM)	visual, auditory	non-wearable	NA	NM	NM
Kim et al. (2020) [35]	pressure sensors (F-scan system, Teckscan Inc., USA)	pressure load	feet	computer monitor (NM)	visual	non-wearable	NA	NM	non-paretic limb, NM, PR and NR
J Jung et al. (2020) [36]	plantar pressure measurement mat with smart socks made of conductive material (GAITRite, CIR System Inc., USA)	pressure load	feet	beam projector and screen (NM)	visual	non-wearable	NA	NM	non-paretic limb, NM, PR and NR
Mottaz et al. (2018) [37]	MRI system (3T Siemens Trio TIM scanner, Siemens Medical Solutions, Germany), EEG system (BioSemi ActiveTwo, BioSemi B. V., Amsterdam, The Netherlands)	realistic head geometry, alpha-band weighted node degree of functional connectivity using absolute imagery part of coherency in the target area (ipsilesional hand motor cortex area, Brodmann area 10)	non-wearable, head	computer screen (NM)	visual	non-wearable	NA	NM	NM, 0.3 s, PR
Boehm et al. (2018) [38]	force plates (NM)	vertical force distribution on each foot	KIINCE	screen, programable plates (NM)	visual, haptic	non-wearable	kinetic immersive interface for neuromuscular coordination enhancement (KIINCE)	NM	non-paretic limb, gait cycle, PR
Tsaih et al. (2018) [39]	EMG system (NM)	EMG signal	tibialis anterior muscle	monitor, speaker (NM)	visual, auditory	non-wearable	NA	NM	maximum voluntary contraction, NM, PR
Schlieβmann et al. (2018) [40]	IMUs (accelerometer, gyroscope, magnetometer; NM)	foot-to-ground angle at heel-strike, stride length, stance duration, swing duration	feet	tablet, speaker or earphones (NM)	visual, auditory	non-wearable, ears	NA	NM, NA	healthy, third stride, PR and NR
Choi et al. (2019) [41]	loading sensor (NM)	user’s weight	center of the metatarsal heads under the paralyzed foot	Speaker (NM)	auditory	foot	NA	NM	non-paretic limb, gait cycle, PR
Lee et al. (2015) [42]	EEG system (QEEG-8 LXE3208, LAXHA Inc., Daejeon, Korea)	sensorimotor rhythm waves amplitude	head	computer screen (NM)	visual	non-wearable	NA	visual	healthy, NM, PR
Hankinson et al. (2022) [43]	IMUs (Mbientlab Inc., San Francisco, CA, USA)	yaw, pitch, roll angles	upper or lower limbs	mobile phone speaker	auditory	non-wearable	NA	NM	baseline, NM, PR and NR
Mihara et al. (2021) [44]	dunctional near-infrared spectroscopy (fNIRS) (OMM-3000, Shimadzu Corp., Tokyo, Japan)	supplementary motor area (SMA) activation	head	screen	visual	non-wearable	NA	NM	NM
Park et al. (2021) [45]	Qualisys Oqus motion capture system	position of reflective markers during swing phase	ankle	screen	visual	non-wearable	treadmill, ankle-foot orthosis	NM	baseline, gait cycle, PR and NR
Skvortsov et al. (2021) [46]	Stadis system (Neurosoft, Ivanovo, Russia)	stance time	ankle	screen	visual	non-wearable	treadmill	NM	baseline, NM, NR

**Table 2 sensors-22-07197-t002:** Clinical specifications (RCT: Randomized Controlled Trial, EG: Experimental Group, CG: Control Group).

Studies	Study Design	Sample Size (EG1, EG2/CG)	Lower Limb Activity	Training Dosing	Evaluation Time Points
Ma et al. (2018) [16]	uncontrolled design	8 (8/0)	walking	7 m-long walkway five times	during procedures
Genthe et al. (2018) [17]	uncontrolled design	9 (9/0)	walking	6 min three times	pre-, post-control and training procedures, 2, 15, 30 min follow-up
Afzal et al. (2019) [18]	uncontrolled design	8 (8/0)	walking	not mentioned	during procedures
Khoo et al. (2017) [19]	non-randomized controlled design	6 (2,2/2)	walking	20 min twice a week for 8 weeks	pre-, mid-, and post-training
Nan et al. (2019) [20]	uncontrolled design	2 (2/0)	sitting	60 min twice a week until 15 sessions	pre-, during, and post-training
Arpa et al. (2019) [21]	RCT	34 (17/17)	sitting	15 min five times a week for 2 weeks	pre-training, immediate post-training, 1-, 3-month follow-up
Tamburella et al. (2019) [22]	randomized cross-over design	10 (5,5/0)	walking	40 min three times a week until 12 sessions	pre- and post-training
Day et al. (2019) [23]	uncontrolled design	10 (10/0)	walking	15 min two sessions at least 3 days apart	during procedures, pre-, and post-training
Guzik et al. (2020) [24]	uncontrolled design	50 (50/0)	walking	20 min five times a week for 2 weeks	pre- and post-training
Surucu et al. (2021) [25]	non-randomized controlled design	40 (20/20)	sitting	20 min five times a week for 3 weeks	pre- and post-training
Ochi et al. (2015) [26]	RCT	26 (13/13)	walking	20 min five times a week for 4 weeks	pre- and post-training
Bae et al. (2020) [27]	RCT	26 (13/13)	sitting	40 min five times a week for 4 weeks	pre- and post-training
Hsu et al. (2019) [28]	uncontrolled design	15 (15/0)	walking	5-min	pre-, early-, late-, and post-training
Givon et al. (2016) [29]	RCT	47 (23/24)	standing	1 h twice a week for 3 months	pre-training, immediate post-training, and 3-month follow-up
Nagano et al. (2020) [30]	uncontrolled design	6 (6/0)	walking	10 min eight sessions over 4 weeks	pre-training, immediate post-training, and 1-month follow-up
Byl et al. (2015) [31]	non-randomized controlled design	12 (7/4)	walking	30 min 12 sessions over 6–8 weeks	pre- and post-training
K-S Jung et al. (2020) [32]	RCT	20 (10/10)	walking	30 min five times a week for 4 weeks	pre- and post-training
Shin et al. (2017) [33]	uncontrolled design	17 (17/0)	walking	5 min	during procedures, post-training
Song et al. (2015) [34]	non-randomized controlled design	40 (20/20)	standing	30 min five times a week for 8 weeks	pre- and post-training
Kim et al. (2020) [35]	RCT	24 (12/12)	walking	30 min three times a week for 4 weeks	pre-training, immediate post-training, and three times per week for 2 weeks follow-up
J Jung et al. (2020) [36]	uncontrolled design	10 (10/0)	walking	not mentioned	during procedures
Mottaz et al. (2018) [37]	uncontrolled design	10 (10/0)	sitting	50 min twice a week over 2 months	pre-training, immediate post-training, and 1-month follow-up
Boehm et al. (2018) [38]	uncontrolled design	10 (10/0)	walking	30-s	during procedures
Tsaih et al. (2018) [39]	RCT	33 (13,11/9)	sitting	40 min 18 sessions over 6 weeks	pre-training, 1-day, 2-week, and 6-week follow-up
Schlieβmann et al. (2018) [40]	uncontrolled design	11 (11/0)	walking	15 min three consecutive sessions	pre-training, immediate post-training, and 4-week follow-up
Choi et al. (2019) [41]	RCT	24 (12/12)	walking	20 min three times a week for 6 weeks	pre- and post-training
Lee et al. (2015) [42]	RCT	20 (20/0)	sitting	30 min three times a week for 8 weeks	pre- and post-training
Hankinson et al. (2022) [43]	RCT	22 (10/12)	upper or lower limb tasks	20 min three times a week for 6 weeks	pre-, mid-, and post-training
Mihara et al. (2021) [44]	RCT	54 (28/26)	sitting	6 min three times a week for 2 weeks	pre-training, immediate post-training, and 2-week follow-up
Park et al. (2021) [45]	uncontrolled design	36	walking	3 min three times	during procedures
Skvortsov et al. (2021) [46]	uncontrolled design	20	walking	18 min eight to eleven times	pre- and post-training

**Table 3 sensors-22-07197-t003:** Outcomes and BS effects on post-stroke recovery: grey indicates non-measured outcomes, blue shows the non-statistically evaluated outcomes, green addresses the condition when at least one measured outcome statistically significantly improved between at least two evaluation time points, dark green corresponds to the condition when at least one measured outcome statistically significantly improved between at least two evaluation time points and the improvement was higher on the experimental group than the control group, and red refers to the condition when none of the measured outcomes demonstrated statistically significant improvements between all evaluation time points.

Studies	Sensor-Based Outcomes				Clinical Outcomes			
	Spatiotemporal	Kinetic	Kinematic	Physiological	Post-Stroke Stage	Motor	Mental	Sensory
Ma et al. [16]	Walking speed Stance time Stride time	Plantar pressure Foot-floor contact area	Peak joint angle					
Genthe et al. [17]	Step length Spatial asymmetry	Peak joint moment Peak AGRF AGRF deficit	Peak joint angle Peak joint angle deficit					
Afzal et al. [18]	Walking speed Time asymmetry							
Khoo et al. [19]	Time asymmetry							
Nan et al. [20]				IAB amplitude		BBS 10-m walk test TUG	HADS MMSE Brief Aphasia Evaluation	
Arpa et al. [21]		Peak joint moment	Joint ROM	Muscle activation		10 m walk test MAS BI		
Tamburella et al. [22]		Human–robot interaction torque				BBS MAS BI MMT FAC Trunk Control Test	QUEST 2.0 VAS of motivation, mood, satisfaction NASA-TLX QCM CESD	
Day et al. [23]	Walking speed Foot clearance		Joint ROM One-dimensional summary of performance from joint angles Performance deficit		FM ESS		MoCA	Proprioception test Start Cancellation Test
Guzik et al. [24]						10 m walk test TUG FIM BI 2 min walk test		
Surucu et al. [25]			Joint ROM	Muscle activation	Brunnstrom neurophysiological assessment	MAS Modified Motor Assessment Scale		
Ochi et al. [26]		Muscle torque			FM	10 m walk test FIM FAC		
Bae et al. [27]	Step length Walking speed Stride length Stance time Swing time Cadence					TUG Tardieu scale WBLT		
Hsu et al. [28]	Step length Walking speed Spatial asymmetry	Peak vertical force		Muscle activation				
Givon et al. [29]	Number of steps walked per day	Grip strength				10 m walk test FIM ARAT FRT IADL questionnaire	Likert-type satisfaction scale Session attendance GDS	
Nagano et al. [30]	Step length Foot clearance Step time Step width		Peak joint angle at MFC					
Byl et al. [31]	Step length		Joint ROM		FM	10 m walk test TUG BBS MMT 6 min walk test TGA 5XSST DGI Number of falls experienced before admitted and during training	VAS of pain	
K-S Jung et al. [32]		Peak vertical force		Muscle activation		TUG TIS		
Shin et al. [33]	Step length Spatial asymmetry Step length deficit						Likert-type satisfaction scale	
Song et al. [34]						10 m walk test TUG LOS test	BDI RCS	
Kim et al. [35]	Step length Walking speed Stride length Spatial asymmetry Single support time Double support time					TUG		
J Jung et al. [36]	Walking speed Stride length Stance time Swing time Time asymmetry Cadence Single support time Double support time Toe-only time Heel-only time							
Mottaz et al. [37]		Grip strength		Alpha-band weighted node degree of functional connectivity	FM	10 m walk test TUG MAS MAL MRC muscle scale 9HPT BBT		
Boehm et al. [38]		Force asymmetry						
Tsaih et al. [39]		Muscle strength				10 m walk test TUG LOS test 6 min walk test		
Schlieβmann et al. [40]	Stride length Swing time		Peak joint angle at heel-strike			10 m walk test TUG MMT WISCI II	QUEST 2.0	PP test
Choi et al. [41]		Centre of pressure				10 m walk test TUG FGA		
Lee et al. [42]	Stance time Cadence	Plantar pressure		SMR waves amplitude		10 m walk test Dual task error		
Hankinson et al. (2022) [43]					FM			
Mihara et al. (2021) [44]	Walking speed				FM	TUG FIM BBS		
Park et al. (2021) [45]	Step length asymmetry		Whole-body angular momentum					
Skvortsov et al. (2021) [46]	Walking speed Foot clearance Stance time Single support time Double support time		Joint angles	Muscle activation		TUG BBS Hauser Ambulance Index (HAI) Standing Balance Test (SBT)		

**Table 4 sensors-22-07197-t004:** Quality assessment results through EPHPP tool for each study.

Studies	Component Ratings						Global Rating
	Selection Bias	Study Design	Confounders	Blinding	Data Collection Methods	Withdrawals and Drop-outs	
Ma et al. [16]	moderate	moderate	strong	strong	strong	weak	moderate
Genthe et al. [17]	weak	moderate	strong	strong	strong	weak	weak
Afzal et al. [18]	weak	moderate	strong	strong	strong	weak	weak
Khoo et al. [19]	moderate	strong	weak	weak	strong	weak	weak
Nan et al. [20]	weak	moderate	strong	strong	strong	weak	weak
Arpa et al. [21]	weak	strong	strong	strong	strong	strong	moderate
Tamburella et al. [22]	moderate	strong	strong	strong	strong	strong	strong
Day et al. [23]	weak	moderate	strong	strong	strong	weak	weak
Guzik et al. [24]	moderate	moderate	strong	strong	strong	weak	moderate
Surucu et al. [25]	moderate	strong	strong	weak	strong	strong	moderate
Ochi et al. [26]	strong	strong	strong	strong	strong	strong	strong
Bae et al. [27]	moderate	strong	strong	weak	strong	strong	moderate
Hsu et al. [28]	weak	moderate	strong	strong	strong	weak	weak
Givon et al. [29]	weak	strong	strong	strong	strong	strong	moderate
Nagano et al. [30]	weak	moderate	strong	strong	strong	weak	weak
Byl et al. [31]	moderate	strong	strong	weak	strong	weak	weak
K-S Jung et al. [32]	weak	strong	strong	strong	strong	weak	weak
Shin et al. [33]	moderate	moderate	strong	strong	strong	weak	moderate
Song et al. [34]	moderate	strong	strong	weak	strong	weak	weak
Kim et al. [35]	weak	strong	strong	strong	strong	strong	moderate
J Jung et al. [36]	moderate	moderate	strong	strong	strong	weak	moderate
Mottaz et al. [37]	weak	moderate	strong	strong	strong	weak	weak
Boehm et al. [38]	weak	moderate	strong	strong	strong	weak	weak
Tsaih et al. [39]	moderate	strong	strong	strong	strong	strong	strong
Schlieβmann et al. [40]	weak	moderate	strong	strong	strong	strong	moderate
Choi et al. [41]	moderate	strong	strong	strong	strong	weak	moderate
Lee et al. [42]	moderate	strong	strong	strong	strong	strong	strong
Hankinson et al. (2022) [43]	strong	strong	weak	strong	strong	strong	moderate
Mihara et al. (2021) [44]	strong	strong	strong	strong	strong	strong	strong
Park et al. (2021) [45]	weak	moderate	strong	weak	strong	strong	weak
Skvortsov et al. (2021) [46]	weak	moderate	strong	weak	strong	strong	weak

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
