# Peer review of "Robotic Biofeedback for Post-Stroke Gait Rehabilitation: A Scoping Review"

_sensors, 2022, doi:10.3390/s22197197_

Round 1

Reviewer 1 Report

1. The formatting of the paper needs attention. The text and headings may have some spaces. 

2. Introduction section needs some revisions. More insight into the previous review papers will provide a clear picture of the need to conduct this review. 

3. More efficient way may be adopted to present the data, especially in the paragraph from lines 325-334, the evaluation strategies can be presented in tabular form. 

4. No proper future recommendations are given as a way forward for researchers working in gait rehabilitation. This limits the efficacy of this review. The authors have claimed in the introduction sections that they provide future directions which are missing. Only statements like "Future research should evaluate clinical sensory effects using statistical tests to power the research conclusions" are mentioned at the end of each paragraph in the discussion section which is not sufficient. The authors should provide a separate heading of future directions and mention all the challenges in detail that could be faced by the researchers while working on gait rehabilitation. 

5. The overall write-up needs major revision. There is no flow between the paragraphs at many points in the manuscript. 

Reviewer 2 Report

Dear Authors

Quite good state-of art, but not very actual. At this time (September 2022) the content should be updated with the latest papers corresponding to this topic. See for example: McMaster, K., Cole, M.H., Chalkley, D. et al. Gait biofeedback training in people with Parkinson’s disease: a pilot study. J NeuroEngineering Rehabil 19, 72 (2022). https://doi.org/10.1186/s12984-022-01051-1

Moreover, there is a number of references related to control, but the details about control strategies are not given in details. It will be very valuable to see the discussion on the controllers (linear, nonlinear, intelligent, adaptive, etc.), their power consumption and quality. It is specially important while the feedback is discussed and sensing is under consideration. Lines 185-205 is a set of references. Such presentation is not acceptable. 

Next real-time aspects are not analyzed in details. After the sentence given in line 185 it is expected that the wide discussion on sensors, type of signals, processing methods, time dependencies, etc. will be given, to show the real-time measurement and control aspects. 

Finally, I'd like to ask you about the robotic based approach, you mentioned the kinetic, kinematic, what about dynamics. What about the torques while training? 

Round 2

Reviewer 1 Report

The authors have sufficently improved the manuscript.